



# Underestimation of oceanic carbon uptake in the Arctic Ocean: Ice melt as predictor of the sea ice carbon pump

Benjamin Richaud[1], Katja Fennel[1], Eric C.J. Oliver[1], Michael D. DeGrandpre[2], Timothée Bourgeois[1, 3], Xianmin Hu[1, 4], and Youyu Lu[4]

[1]Department of Oceanography, Dalhousie University, Halifax, NS, Canada
[2]Department of Chemistry and Biochemistry, University of Montana, Missoula, MT, USA
[3]NORCE Norwegian Research Centre, Bjerknes Centre for Climate Research, Bergen, Norway
[4]Bedford Institute of Oceanography, Department of Fisheries and Oceans, Dartmouth, NS, Canada

**Correspondence:** Benjamin Richaud (benjamin.richaud@dal.ca)

**Abstract.** The Arctic Ocean is generally undersaturated in $CO_2$ and acts as a net sink of atmospheric $CO_2$. This oceanic uptake is strongly modulated by sea ice, which can prevent air-sea gas exchange and has major impacts on stratification and primary production. Moreover, carbon is stored in sea ice with a ratio of alkalinity to dissolved inorganic carbon that is larger than in seawater. It has been suggested that this storage amplifies the seasonal cycle of seawater $pCO_2$ and leads to
an increase in oceanic carbon uptake in seasonally ice-covered regions compared to those that are ice-free. Given the rapidly changing ice-scape in the Arctic Ocean, a better understanding of the link between the seasonal cycle of sea ice and oceanic uptake of $CO_2$ is needed. Here, we investigate how the storage of carbon in sea ice affects the air-sea $CO_2$ flux and quantify its dependence on the ratio of alkalinity to inorganic carbon in ice. To this end, we present two independent approaches: a theoretical framework that provides an analytical expression of the amplification of carbon uptake in seasonally ice-covered
oceans, and a simple parameterization of carbon storage in sea ice implemented in a 1D physical-biogeochemical ocean model. Sensitivity simulations show a linear relation between ice melt and the amplification of seasonal carbon uptake. A 30 % increase in carbon uptake in the Arctic Ocean is estimated compared to ice melt without amplification. Applying this relationship to different future scenarios from an Earth System Model that does not account for the effect of carbon storage in sea ice suggests that Arctic Ocean carbon uptake is underestimated by 5 to 15 % in these simulations.

## 1 Introduction

According to current estimates, the Arctic Ocean accounts for 5 to 14 % of the total global oceanic carbon uptake (Bates and Mathis, 2009; Schuster et al., 2013; MacGilchrist et al., 2014; Yasunaka et al., 2016, 2018). Longer open water seasons are expected to increase Arctic oceanic carbon uptake in the near term, with complex feedbacks altered by climate change (Lannuzel et al., 2020; Steiner et al., 2015; Ouyang et al., 2020), but the scarcity of biogeochemical observations in the Arctic
Ocean prevents reliable calculations of carbon flux (e.g., Landschützer et al., 2014), as well as proper validation of climate models in the region.





In the Arctic Ocean, air-sea gas exchange is mostly prevented by sea ice in winter while being partially allowed in summer when there is open water. While carbon fluxes between ice and atmosphere are known to exist (Delille, 2006; Miller et al., 2011; Geilfus et al., 2012; Nomura et al., 2010), large uncertainties remain on their significance (Watts et al., 2022) and sea
ice is therefore often considered as a physical lid. Melting and freezing of sea ice affect the partial pressure of $CO_2$ ($pCO_2$) in the surface ocean and thus the air-sea flux which depends on the $pCO_2$ gradient between the surface ocean and overlying atmosphere (e.g., Wanninkhof, 2014). Melting dilutes dissolved constituents in the surface ocean, thus decreasing dissolved inorganic carbon (DIC=$[CO_2] + [HCO_3^-] + [CO_3^{2-}]$) and $pCO_2$; the opposite is true when ice is forming (DeGrandpre et al., 2019). Moreover, when sea ice forms, it rejects the dissolved salts in the brine filling the gaps between the crystal lattice. Part
of this salty, carbon-rich brine is expelled from the ice (Miller et al., 2011). Sinking of some of this dense brine provides a pathway for carbon export below the mixed layer (König et al., 2018; Barthélemy et al., 2015, and references therein). DIC and alkalinity (here simplified as carbonate alkalinity = $[HCO_3^-] + 2[CO_3^{2-}]$) are also stored inside the sea ice in brine channels. Since alkalinity is retained preferentially (Rysgaard et al., 2007, 2009), this carbon storage in ice affects surface ocean $pCO_2$ during melting and freezing beyond the above-mentioned effects of dilution-, concentration-, and brine-driven carbon export.

During ice growth, precipitation of ikaite (hydrated $CaCO_3$) occurs within sea ice: $Ca^{2+} + 2HCO_3^- \rightarrow CaCO_{3(s)} + H_2O + CO_{2(aq)}$ (Dieckmann et al., 2008). This precipitation traps alkalinity inside the ice crystal lattice and increases DIC in the brine (Rysgaard et al., 2009). Brine drainage then expels part of this DIC, lowering its concentration inside the sea ice while increasing it in underlying water. Brine drainage also allows for the exchange of nutrients between ice and ocean, feeding sympagic (ice-affiliated) ice algae in spring and further decreasing DIC in ice through primary production (Vancoppenolle
et al., 2013). By the end of the ice growth season, the alkalinity to DIC ratio is significantly higher in sea ice than in adjacent seawater. During the melt season, ikaite dissolves in seawater preferentially releasing alkalinity over DIC, thus further lowering sea surface $pCO_2$ and increasing oceanic carbon uptake. This process is commonly referred to as the "sea-ice carbon pump" (Rysgaard et al., 2007). The intensity of this pump and the underlying drivers are still subject to discussion (e.g. Delille et al., 2014).

While the role of biotic and abiotic processes on the carbon cycle within sea ice is becoming better understood, their impact on the underlying seawater is less clear. Using a conceptual model, Rysgaard et al. (2011) estimated that the sea-ice carbon pump could generate an additional uptake of 50 TgC yr$^{-1}$, accounting for 17 to 42 % of high latitude carbon uptake. Applying an empirical relationship between $CO_2$ flux and sea ice temperature to a numerical model, Delille et al. (2014) estimate that Antarctic sea ice uptakes 29 TgC yr$^{-1}$. In their idealized climate scenarios, Moreau et al. (2016) found that the impact of
carbon storage in sea ice weakens the Arctic $CO_2$ sink while Grimm et al. (2016) suggested a moderate role of the sea-ice carbon pump in the modern global carbon cycle but acknowledged its potential importance on regional scales. Finally, in a regional ocean model, Mortenson et al. (2020) showed that the amplitude of the DIC seasonal cycle increased by 25 % in the surface ocean but with an unchanged annual carbon uptake (<1 % increase). The discrepancies between those studies suggest that the importance of carbon storage in ice in the global carbon cycle is still an open question, with increasing relevance due
to the current and projected evolution of sea ice.



Arctic sea ice extent and thickness have declined rapidly over the past decades at a rate of -83,000 km$^2$ yr$^{-1}$ for September ice extent during the 1979-2018 period and with a decline in ice thickness by 65 % from 1975 to 2012 (Meredith et al., 2019). This decline is expected to continue. Arctic amplification, a combination of positive feedbacks including summer albedo loss and changes in cloudiness, is leading to twice the rate of warming of the atmosphere compared to the global average (Meredith

et al., 2019, Box 3.1). Increased "Atlantification" of the Eurasian Arctic Basin, characterized by a progression of Atlantic water masses into the Arctic seas, is contributing to amplified basal ice melt (Polyakov et al., 2017). These dynamic and thermodynamic processes have direct impacts on sea ice seasonality and extent (Perovich and Richter-Menge, 2009) and ice-free summers are predicted to happen within the next few decades (Overland and Wang, 2013; Notz and Community, 2020). Yet, since sea ice extent in winter decreases slower than in summer, the seasonally ice-covered area is expanding. Such an

amplified seasonality in sea ice may intensify the sea ice carbon pump, as sea ice forms in open water that had previously been perennially ice-covered.

We use two independent and complementary approaches to investigate the supplementary carbon flux in the Arctic Ocean. We define the supplementary carbon flux $\Delta\mathcal{F}$ as the fraction of the air-sea CO$_2$ flux that is solely due to the storage of carbon and alkalinity in ice. This term is quantified here as the difference in air-sea CO$_2$ flux between a situation where there is no ice-

ocean carbon flux, and situations where ice-ocean carbon flux occurs. First, we combine a set of mathematical formulations to obtain an equation that provides a theoretical framework for the description of the impact of alkalinity and DIC storage in sea ice on air-sea CO$_2$ fluxes. These theoretical considerations suggest that sea-ice melt and open-water fraction are the main drivers of an increased oceanic carbon uptake induced by storage of alkalinity and DIC in sea ice. Second, a simple parameterization of the presence of alkalinity and DIC inside the sea ice is implemented in a one-dimensional (1D) ocean

model applied to different locations of the Arctic. A large set of sensitivity runs with this 1D model consolidates and expands on the role and importance of melt and open-water-fraction and shows that the alkalinity-to-DIC ratio in sea ice plays a major role in the magnitude of the increased uptake. By forcing the model with a wide range of plausible ice conditions, we obtain a predictive linear relationship between annual ice melt and ice-induced annual supplementary carbon uptake ($\Delta\mathcal{F}$). This relationship can be used to correct carbon uptake estimates from numerical models that do not account for carbon storage in

ice. By applying the relationship to an Earth System Model (ESM) from the sixth phase of the Climate Model Intercomparison Project (CMIP6) ensemble, we show how the impact of sea ice on carbon uptake may evolve under different future emission scenarios.

## 2  Theoretical Framework for Ice-Sea Carbon Flux and Induced Air-Sea CO$_2$ Uptake

The impact of carbon storage in sea ice on the air-sea CO$_2$ flux is analyzed using differential equations that account for the

impact of freezing and melting on surface water alkalinity and DIC. The air-sea flux is expressed as a function of sea surface $p$CO$_2$, which depends on temperature, salinity, alkalinity and DIC.

We assume the flux of alkalinity and DIC between the sea ice and the underlying water to be proportional to the freshwater flux induced by freezing and melting of sea ice, $\mathcal{F}_{FW}^{ice-sea}$ (m s$^{-1}$), and the concentration of alkalinity and DIC inside the ice.



The DIC and alkalinity concentrations are assumed to be homogeneous in the ice. The freshwater flux is positive (downward)

for melting. The change in sea surface $pCO_2$, written $\frac{\partial pCO_2^{ice-sea}}{\partial t}$, resulting from the freshwater flux can then be expressed as

$$\frac{\partial pCO_2^{ice-sea}}{\partial t}(t) = \frac{1}{H_0} g(t) \mathcal{F}_{FW}^{ice-sea}(t)$$

with

$$g(t) = \frac{\partial pCO_2}{\partial Alk}(t)[Alk]_{ice} + \frac{\partial pCO_2}{\partial DIC}(t)[DIC]_{ice} \tag{1}$$

where $H_0$ is the mixed layer depth (in m), considered constant for ease of interpretation; $[Alk]_{ice}$ and $[DIC]_{ice}$ are the

concentrations of alkalinity and DIC inside sea ice (mmol m$^{-3}$) and $\frac{\partial pCO_2}{\partial Alk}$ and $\frac{\partial pCO_2}{\partial DIC}$ are the fractional change of $pCO_2$

with alkalinity and DIC, respectively (µatm m$^3$ mmol$^{-1}$). Note that $\frac{\partial pCO_2}{\partial Alk}$ and $\frac{\partial pCO_2}{\partial DIC}$ are generally non-linear.

The relation between the air-sea flux of CO$_2$ and seawater $pCO_2$ is

$$\mathcal{F}_{CO_2}^{air-sea} = k_g S_{CO_2} \lambda (pCO_2^{atm} - pCO_2) \tag{2}$$

where $pCO_2$ and $pCO_2^{atm}$ refer to $pCO_2$ in surface seawater and atmosphere (µatm) resp., $k_g$ is the gas transfer velocity (m

s$^{-1}$), $S_{CO_2}$ is the CO$_2$ solubility (mol m$^{-3}$ µatm$^{-1}$) and $\lambda$ is the fraction of open water (lead fraction, unitless; Ahmed et al.,

2019). Here, the air-sea CO$_2$ flux is defined as positive downward.

The supplementary flux, $\Delta \mathcal{F}_t$, is calculated as the difference between a case with carbon storage in ice, referred to as ICE,

and a control (CTRL) case, where storage is not considered, i.e.

$$\Delta \mathcal{F}_t = \mathcal{F}_{CO_2}^{air-sea, \, ICE} - \mathcal{F}_{CO_2}^{air-sea, \, CTRL} = -k_g S_{CO_2} \lambda (pCO_2^{ICE} - pCO_2^{CTRL})$$

with $pCO_2^{ICE}$ and $pCO_2^{CTRL}$ the sea surface $pCO_2$ in the ICE and CTRL cases, respectively. In the rest of this manuscript,

we will denote $\Delta pCO_2^{i-c} = pCO_2^{ICE} - pCO_2^{CTRL}$.

We assume that in both CTRL and ICE cases, sea surface $pCO_2$ experiences the same alterations due to biological processes

and changes in temperature and salinity caused by vertical and horizontal mixing and air-sea-ice interactions. This assumption

neglects the possibility that non-linearities of the carbonate system lead to differences in the impact of these processes on $pCO_2$

between the CTRL and ICE cases. Moreover, we assume that $\frac{\partial pCO_2}{\partial DIC}$ is constant. Calculations conducted with CO2SYS (Lewis

and Wallace, 1998) based on mooring data located in the Beaufort Gyre (DeGrandpre et al., 2019, 78° N, 150° W) and our

model data (see Beaufort Gyre setup in Sect. 3.1) yield a coefficient of variation of $\frac{\partial pCO_2}{\partial DIC}$ of only 6 % and 5 %, respectively.

This supports the assumption of a constant $\frac{\partial pCO_2}{\partial DIC}$ over the range of expected DIC. These two assumptions are only used in

this theoretical derivation, not in the numerical analysis.

The change of $pCO_2$ over time can be written as

$$\frac{\partial pCO_2}{\partial t} = \frac{\partial pCO_2}{\partial DIC}\frac{\partial DIC}{\partial t} + \frac{\partial pCO_2}{\partial Alk}\frac{\partial Alk}{\partial t} + \frac{\partial pCO_2}{\partial T}\frac{\partial T}{\partial t} + \frac{\partial pCO_2}{\partial S}\frac{\partial S}{\partial t}$$

with the temperature and salinity contributions (the last two terms on the right-hand side) being identical in the ICE and CTRL

cases.



The contributions from alkalinity and DIC can come from advection, diffusion, mixing, biological processes (production,
respiration, remineralization), and air-sea or ice-ocean carbon fluxes. As already described, the ice-ocean carbon flux modifies
the surface seawater $pCO_2$, which in turn impacts the air-sea carbon flux. Here, the ice-ocean and air-sea carbon fluxes are the
two only processes that are not considered as identical between CTRL and ICE cases and are therefore the only two terms left
when subtracting the equations for $\frac{\partial pCO_2}{\partial t}$ for the CTRL and ICE cases from each other. The following differential equation
governing the evolution of $\Delta pCO_2^{i-c}$ can be derived (see details in the supplement):

$$\frac{\partial \Delta pCO_2^{i-c}}{\partial t}(t) = -\frac{\partial pCO_2}{\partial DIC}\frac{1}{H_0}k_g(t)S_{CO_2}(t)\lambda(t)\Delta pCO_2^{i-c}(t) + \frac{1}{H_0}g(t)\mathcal{F}_{FW}^{ice-sea}(t) \tag{3}$$

The solution to Eq. 3 is:

$$\Delta pCO_2^{i-c}(t) = e^{-A(t)}\int_0^t \frac{1}{H_0}g(s)\mathcal{F}_{FW}^{ice-sea}(s)e^{A(s)}ds \tag{4}$$

where $A(t)$ is a primitive of $\frac{\partial pCO_2}{\partial t}\frac{1}{H_0}k_gS_{CO_2}\lambda$ and $s$ is the variable of integration, with units of seconds. The primitive of
a function can be calculated as its time integral plus an unknown constant $\alpha$

$$A(t) = \int_0^t \frac{\partial pCO_2}{\partial DIC}\frac{1}{H_0}k_g(s)S_{CO_2}(s)\lambda(s)ds + \alpha$$

This yields a solution for the instantaneous difference in $pCO_2$ between CTRL and ICE scenarios. To retrieve the previously
defined supplementary carbon uptake, i.e., the cumulative air-sea $CO_2$ flux that is induced by the $pCO_2$ change, we can insert
Eq. 4 into the left-hand side of Eq. 3 and integrate over a period $T$:

$$\Delta\mathcal{F}_t(T) = \frac{1}{\frac{\partial pCO_2}{\partial DIC}}\int_0^T g(t)\mathcal{F}_{FW}^{ice-sea}(t)\left(e^{A(t)-A(T)} - 1\right)dt \tag{5}$$

A unique derivation to our knowledge, this formulation is composed of three main terms: $g(t)$, which is a function of the
concentration of alkalinity and DIC in the ice (Eq. 1); the freezing-melting flux $\mathcal{F}_{FW}^{ice-sea}$; and the more complicated exponential
of the primitive, which contains the lead fraction $\lambda$ in $A(t)$. $A(t)$ is an integral of the lead fraction and can be interpreted as
keeping a memory of the evolution of the ice conditions.

The sign of $g(t)$ determines the sign of $\Delta\mathcal{F}_t$. Using realistic alkalinity and DIC values for the Arctic Ocean (e.g. $[Alk]_{sw}$
= 2300 mmol m$^{-3}$, $[DIC]_{sw}$ = 2100 mmol.m$^{-3}$, $[Alk]_{ice}$ = 540 mmol m$^{-3}$ and $[DIC]_{ice}$ = 300 mmol m$^{-3}$, as in Rysgaard
et al. (2011); Miller et al. (2014) or $[Alk]_{ice}$ = 415 mmol m$^{-3}$ and $[DIC]_{ice}$ = 330 mmol m$^{-3}$, as in DeGrandpre et al. (2019);
and Revelle and alkalinity factors of 14 and -13.3 respectively, as in Takahashi et al. (1993)) yields a negative sign for $g(t)$. It
can be shown that the term between parentheses in Eq. 5 is always negative, meaning that for ice melt ($\mathcal{F}_{FW}^{ice-ocean} > 0$), $\Delta\mathcal{F}_t$
is downward (uptake); the opposite is true for ice formation (cf. Supplementary materials). According to this formulation, the
dependency of $\Delta\mathcal{F}_t$ on $[Alk]_{ice}$ and $[DIC]_{ice}$ is bi-linear due to the shape of $g(t)$.





It is important to note that the gas transfer velocity and the $CO_2$ solubility, used in the primitive $A(t)$, require no assumption of shape or value. The gas transfer velocity $k_g$ can depend on the wind speed (e.g. Wanninkhof, 2014), on the wave slope (Bogucki et al., 2010) or on turbulence generated by ice drag and convection (Loose et al., 2014). Similarly, the $CO_2$ solubility could follow Weiss (1974) or any other expression.

One can calculate the solution numerically using the carbonate properties of seawater and sea ice (i.e., their alkalinity and DIC), the sea ice concentration, the ice-ocean freshwater flux, the gas transfer velocity (e.g., using Loose et al. (2014)) and the $CO_2$ solubility (which depends on temperature and salinity, Weiss, 1974). The product of the Schmidt number and $CO_2$ solubility can reasonably be considered constant (Etcheto and Merlivat, 1988), therefore removing the dependency on temperature and salinity Wanninkhof (2014, their Eq. 6) and providing an even simpler form than proposed above.

In order to interpret the role of $\lambda$, its value can be constrained as follow. We assume that ice formation is associated with full ice cover ($\lambda \approx 0$) and that melting occurs in open waters ($\lambda \approx 1$). We will see that this is supported by ocean model output in Sect. 4.1. Then, during ice formation when $\lambda$ is very small, $\lim_{\lambda \to 0}(e^{A(t)-A(T)} - 1) = 0$.

This implies that the integrand in Eq. 5 is negligible during freezing and non-negligible during melting. Thus, ice formation has a relatively small contribution to the temporal integral of the supplementary carbon flux, while ice melting significantly 160   increases the $CO_2$ flux. Since melting leads to uptake, according to the sign examination above, the net outcome of the supplementary carbon flux is uptake.

Note that if $H_0$ was assumed to be variable in time, it would remain inside the integrands on both sides of Eq. 5. The integrand is then likely to be small during the melting season when the mixed layer shoals, and larger during the freezing season, when $\lambda$ is close to 0 and the integrand is already small. A variable mixed layer depth would therefore reinforce the 165   already dominant influence of the melting season in the value of the supplementary carbon uptake.

## 3   Numerical ocean model

We implemented a parametrization of carbon storage and release by sea ice in a 1D ocean model, independent of the theoretical arguments in Sect. 2, to investigate its impact on the air-sea $CO_2$ flux in different regions of the Arctic Ocean. We do not use any of the results or assumptions from Section 2. By using a wide range of initial and forcing conditions derived from a realistic 170   3D model, a large ensemble of 1D simulations is generated to account for spatial and temporal variability in forcing conditions. Analysis of the ensemble provides insights into the main drivers of the supplementary carbon uptake and allows us to derive a formula to estimate the supplementary carbon flux in existing Earth System Model (ESM) simulations. Here we describe the 1D model set-up and forcings, as well as the ESM outputs used to project the evolution of the supplementary carbon flux in different scenarios.

### 3.1   One-dimensional ocean model

The 1D General Ocean Turbulence Model (GOTM, Burchard et al., 1999) is coupled to the Pelagic Interactions Scheme for Carbon and Ecosystem Studies volume 2 (PISCES-v2, Aumont et al., 2015), specifically adapted to the Framework for



Aquatic Biogeochemical Models (FABM, Bruggeman and Bolding, 2014). The vertical grid has fixed layer thicknesses, with a resolution of 1 m near the surface and increasing with depth (9 layers in the first 10 m, 24 layers in the first 100 m). Air-sea

$CO_2$ flux is calculated by the model using values from Wanninkhof (2014) with 10 m wind speed. The carbonate chemistry in the model follows the OCMIP protocols (Orr, 1999).

The evolution of alkalinity and DIC in surface waters is parameterized by

$$\frac{dDIC}{dt} = [DIC]_{ice}\frac{\mathcal{F}^{ice-sea}_{FW}}{H_{cell}} + \frac{\mathcal{F}^{air-sea}_{CO_2}}{H_{cell}} + Phys_{DIC} + Bio_{DIC}$$

$$\frac{dAlk}{dt} = [Alk]_{ice}\frac{\mathcal{F}^{ice-sea}_{FW}}{H_{cell}} + Phys_{Alk} + Bio_{Alk}$$

where $[Alk]_{ice}$ and $[DIC]_{ice}$ are the Alkalinity and DIC concentrations in ice and held constant throughout a simulation, $\mathcal{F}^{ice-sea}_{FW}$ is the flux of freshwater between ice and ocean due to ice melt or freezing (m s$^{-1}$, positive downward), $H_{cell}$ is the thickness of the uppermost ocean grid cell (here 1.02 m), $Phys$ includes the advective and dispersive transport terms and $Bio$ represents the biological sources and sinks of Alk and DIC, and $\mathcal{F}^{air-sea}_{CO_2}$ is the air-sea $CO_2$ flux. Preliminary runs showed that the biological terms have a negligible impact on supplementary carbon uptake (less than 1 % normalized difference) and

were therefore deactivated for the ensemble runs to save computational effort.

Surface forcings were prescribed from a 3D physical-biogeochemical-ice-ocean model based on NEMO-LIM-PISCES (Madec et al., 2017; Rousset et al., 2015; Aumont et al., 2015) for the North Atlantic, North Pacific, and Arctic Oceans, hereafter referred to as the NAPA model. The NAPA model, including the validation with observational data, is documented in Zhang et al. (2020) and Zheng et al. (2021). In our application of this 3D model, the atmospheric forcing was obtained from

ERA-5 reanalysis product (Hersbach et al., 2020) from 2014 to 2019. Outputs were written out daily, providing the necessary temporal resolution to capture sub-seasonal variability. We used the simulated ice concentration, latent and sensible heat fluxes, longwave and shortwave radiative fluxes, freshwater fluxes (due to ice melt-freeze and evaporation-precipitation), and momentum fluxes (due to wind and ice drift) at the top of the surface layer of the model, calculated as a weighted average between open water and under ice conditions to force the 1D model. This methodology allows us to simulate the impact of sea ice in

our 1D model without having to resort to a full ice component. Other inputs necessary for air-sea $CO_2$ flux include the wind speed and mean sea-level pressure from ERA5, as well as atmospheric $pCO_2$ from the Alert Station, Northwest Territories of Canada (Keeling et al., 2001).

In generating the ensemble of 1D simulations, every 10th horizontal grid cell of the NAPA domain was selected with the following exceptions. Since our focus is on open-ocean conditions with a significant presence of sea ice, coastal locations with

water depths shallower than 100 m as well as the Canadian Arctic Archipelago, Hudson Bay and the Baltic Sea were excluded. Also excluded were grid cells with both ice melt and freezing rates of less than 0.1 m yr$^{-1}$. Given NAPA's average grid spacing of  12 km in the Arctic, every 10th grid cell leads to roughly one cell every 120 km for a total of 732 cells covering a wide range of ice conditions. For each of these locations, we ran the 1D model for six 1-year simulations starting on January 1st for the years 2014 to 2019, with initial conditions from the NAPA model. Since the 1D model cannot explicitly represent horizontal





**Table 1.** Description of 1D model runs. For more in-depth sensitivity experiments (*), we selected a representative station in the Beaufort Gyre (78° N, 150° W) for the years 2014 and 2015, where mooring observations are available (DeGrandpre et al., 2019, see comparison in the supplementary materials).

| Acronym: Description | Alk in ice (mmol m$^{-3}$) | DIC in ice (mmol m$^{-3}$) | Alk:DIC ratio |
|---|---|---|---|
| CTRL: Simulation without carbon storage in ice | 0 | 0 | N/A |
| ICE: Simulation with carbon storage in ice | 540 | 300 | 1.80 |
| ICE2*: Simulation with carbon storage in ice | 415 | 330 | 1.26 |
| Sensitivity*: Simulation with carbon storage in ice | 340 to 700 with a 20 increment | 260 to 600 with a 20 increment | 0.57 to 2.69 |

advection, its solutions were nudged toward the properties simulated by the NAPA model with a timescale of 4 months for temperature and salinity and 1 year for alkalinity and DIC.

Based on the above setup, we systematically ran the 1D model in two configurations CTRL (no carbon in sea ice) and ICE (storage of carbon in sea ice). The runs are listed in Table 1. The supplementary carbon uptake $\Delta\mathcal{F}_m$ is calculated as the difference in annual air-sea $CO_2$ flux between the ICE (or ICE2) and the CTRL runs. We consider the air-sea CO2 flux in the

CTRL run as the baseline, since it corresponds to the values reported by numerical models that do not account for the sea ice carbon pump. Potential predictors of the supplementary carbon flux are investigated including the net freezing-melting flux (the integral over a year of the freshwater flux between ice and ocean), the gross melting (freezing) flux which only accounts for ice melt (formation), and the yearly integrated ice concentration (which ranges between 0 and 365). We bin the latter metric into 9 bins of equal size and applied a linear regression between gross annual ice melt and the supplementary carbon uptake

for each of these bins, which can be considered different ice regimes.

### 3.2 Application to an Earth System Model

ESM output from the CMIP6 suite of models can be used to estimate the supplementary carbon flux in projected future climate scenarios. We chose the ACCESS-ESM1.5 model (Ziehn et al., 2020) because it has a plausible simulation of sea ice (according to Notz and Community, 2020) and its monthly-averaged freshwater ice-ocean flux due to ice thermodynamics

(CF standard name: *fsitherm*) and air-sea CO2 flux (CF standard name: *fgco2*) are available. The horizontal resolution of the ocean component of ACCESS-ESM1.5 is 1°, with 50 vertical levels. The historical simulation covers 1850 to 2015, and three





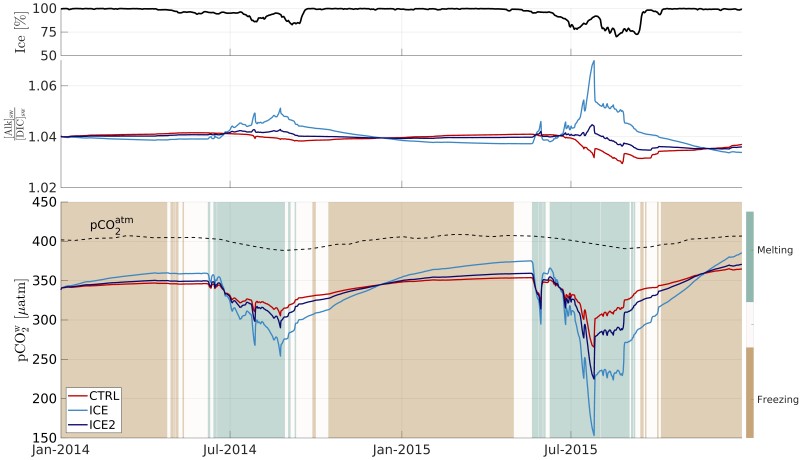

**Figure 1.** Model outputs for a grid cell representative of Central Beaufort Gyre (78° N, 150° W) over 2014-2015. (a): Ice concentration. (b): Surface seawater alkalinity to DIC ratio for the CTRL (no ice carbon flux; red line), ICE ($[Alk]_{ice}$ = 540 mmol m$^{-3}$, $[DIC]_{ice}$ = 300 mmol m$^{-3}$; light blue line) and ICE2 ($[Alk]_{ice}$ = 415 mmol m$^{-3}$, $[DIC]_{ice}$ = 330 mmol m$^{-3}$; dark blue line) runs. (c): Ice melt and formation (>3mm day$^{-1}$; background color); observed atmospheric $p$CO$_2$ at the Alert weather station (dashed black line) and simulated surface seawater $p$CO$_2$ (solid lines) for the three above-mentioned runs.

available Shared Socio-economic Pathways (SSP) scenarios (SSP1-2.6, SSP2-4.5, and SSP5-8.5) cover the period from 2015 to 2100. Monthly outputs of the freshwater ice-ocean flux and air-sea CO$_2$ flux were extracted for the historical simulation and the three SSP scenarios.

Consistent with the methodology applied to our 1D model study, only grid cells where ice melt or freeze was over 0.1 m yr$^{-1}$ were used. For each year between 1850 and 2100 and each remaining grid cell, ice-ocean freshwater flux was summed for melting months only (thus excluding negative values of *fsitherm*).

# 4    Results

## 4.1    Ensemble of 1D model experiments

In the CTRL run (no ice-ocean carbon flux), $p$CO$_2$ increases to maxima of 347 and 354 µatm in the winters of 2014 and 2015, respectively, due to the removal of freshwater and associated concentration of DIC and alkalinity (Fig. 1c). The $p$CO$_2$ decreases to minima of 305 and 265 µatm in the summers of 2014 and 2015, respectively, when ice melts and dilutes seawater constituents. In the ICE run, when accounting for the ice-ocean carbon flux, the seasonal cycle of $p$CO$_2$ is similar, but amplified reaching higher maxima (360 and 375 µatm in 2014 and 2015, respectively) and lower minima (254 and 153 µatm in 2014 and
2015, respectively).




The reason for this amplification is illustrated in Fig. 1b. When accounting for the ice-ocean carbon flux, the alkalinity-to-DIC ratio at the surface decreases during the freezing season and increases during the melting season, a behavior that is opposite to the control run. Since an increase in alkalinity decreases $pCO_2$ and an increase in DIC increases $pCO_2$, the storage and release of both properties by sea ice have counteracting effects. The alkalinity effect dominates and leads to a decrease in

seawater $pCO_2$ when ice melts, amplifying the seasonal cycle of $pCO_2$. The degree of amplification depends on the values of $[Alk]_{ice}$ and $[DIC]_{ice}$, as illustrated by comparing the ICE and ICE2 runs with different alkalinity-to-DIC ratios of the ice-ocean carbon flux. ICE2, which has a lower alkalinity-to-DIC ratio (1.26 compared to 1.80 for ICE), shows lower maximum values (350 and 359 µatm in 2014 and 2015, respectively) and higher minimum values (290 and 225 µatm in 2014 and 2015, respectively) of $pCO_2$, compared to ICE.

How this amplification of the seasonal cycle of $pCO_2$ affects the seasonal air-sea $CO_2$ flux depends on the ice cover shown in Fig. 1a. According to the formulation in Eq. 2, almost complete ice cover ($\lambda = 0$) in winter results in an air-sea $CO_2$ flux close to 0 when $pCO_2$ is highest. Lower sea ice cover in summer allows for some air-sea gas exchange directly proportional to the air-sea $pCO_2$ gradient. Integrated over a full seasonal cycle, the amplification of the $pCO_2$ cycle results in net oceanic $CO_2$ uptake added to the baseline. In the case of the Beaufort Gyre station location, averaged over both years, this supplementary

uptake $\Delta\mathcal{F}_m$ amounts to 45.5 mmol C m$^{-2}$ yr$^{-1}$ for an alkalinity-to-DIC ratio of 1.80 (ICE) and 13.6 mmol C m$^{-2}$ yr$^{-1}$ for a ratio of 1.26 (ICE2), over a 3-fold difference. Note that these are low flux values relative to other oceans (usually higher than 1 mol C m$^{-2}$ yr$^{-1}$), mainly because of the ice cover.

The effect of carbon storage on the annual net $CO_2$ flux is explored more thoroughly for the Beaufort Gyre location by varying $[Alk]_{ice}$ from 340 to 700 mmol eq m$^{-3}$ and $[DIC]_{ice}$ from 260 to 600 mmol m$^{-3}$ (Fig. 2). The net $CO_2$ flux (Fig. 2a)

and supplementary carbon uptake $\Delta\mathcal{F}_m$ (Fig. 2b) are strongly dependent on the alkalinity-to-DIC ratio in ice (white contours). Notably, the net $CO_2$ flux varies by a factor of 2 to 3 for realistic values of the alkalinity-to-DIC ratio. Thus, alkalinity and carbon storage in ice has a significant impact on the net air-sea $CO_2$ flux in the model.

Next, we investigate the role of ice conditions, including the freezing-melting rate and ice concentration on the air-sea $CO_2$ flux. The NAPA model simulates a wide range of ice melt rates over the Arctic Ocean, spanning from 0 to over 7 m yr$^{-1}$ and

with areas of high ice melt in the Labrador and East Greenland Currents and the southern edge of the Beaufort Gyre (Fig. 3a). The NAPA model also simulates freezing conditions that mostly occur when the lead fraction is close to 0 (Supplementary materials, Fig. S2). Indeed, over 88 % of the freezing days occur when the ice concentration is above 0.9. This supports the assumption made in Sect. 2, where we considered freezing to mostly occur when the lead fraction is close to 0.

Gross freezing rates and yearly integrated ice coverage are poorly correlated to $\Delta\mathcal{F}_m$ ($r^2$=0.12 and $r^2$=0.15 respectively).

Yearly net freezing-melting is more strongly correlated with $\Delta\mathcal{F}_m$ ($r^2$=0.39). However, a better predictor of $\Delta\mathcal{F}_m$ is ice melt, excluding any freezing, hereafter called gross annual melt ($r^2$=0.86; Fig. 4). An explanation for this strong relation is that winter ice cover prevents air-sea flux during the freezing period. This is an independent confirmation of the interpretation of the mathematical derivation made in Sect. 2.





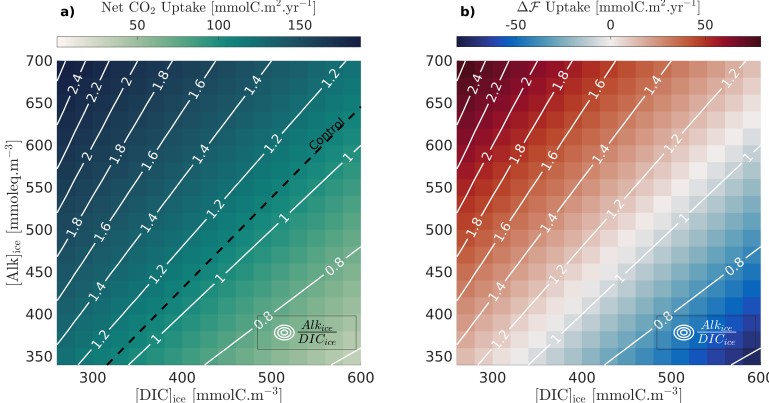

**Figure 2.** Dependence of annual net $CO_2$ uptake on alkalinity and DIC concentrations in ice. The two panels show results from model sensitivity runs for a wide range of alkalinity and DIC values in ice (20 units increments). (a): Annual net $CO_2$ uptake (background colors); net $CO_2$ uptake value for the standard run is highlighted by the dashed black line for reference (note that the standard run is not part of the runs shown in the background colors, since $[Alk]_{ice}=[DIC]_{ice}=0$). (b): Supplementary carbon flux $\Delta\mathcal{F}_m$ due to carbon storage in ice (background colors). ALK:DIC ratio in ice is superimposed (white lines).

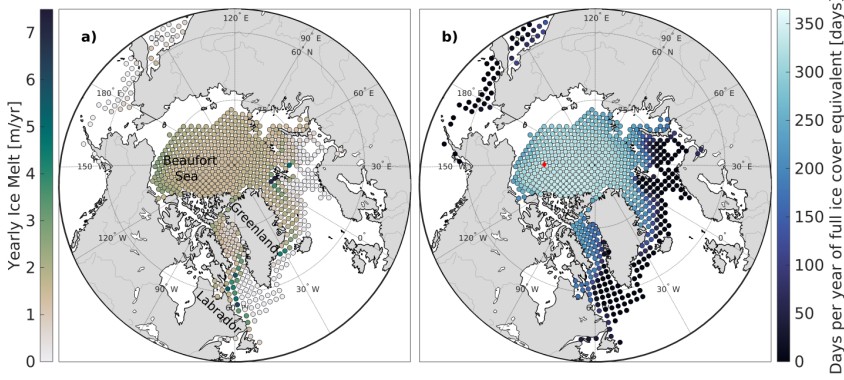

**Figure 3.** Region of interest and sea ice regime from the NAPA model domain. Each dot gives the location of the forcing conditions used to force the 1D model in this study. (a): Mean gross annual ice melt. (b): Mean yearly temporal integral of ice concentration. The red dot shows the grid cell used for Fig. 1 and 2.

The high correlation between the gross annual ice melt ($\mathcal{F}_{Melt}$) and $\Delta\mathcal{F}_m$ gives confidence in a linear model relating those

two metrics:

$$\Delta\mathcal{F}_m = 113.6 \cdot \mathcal{F}_{Melt} - 10.1 \qquad (6)$$





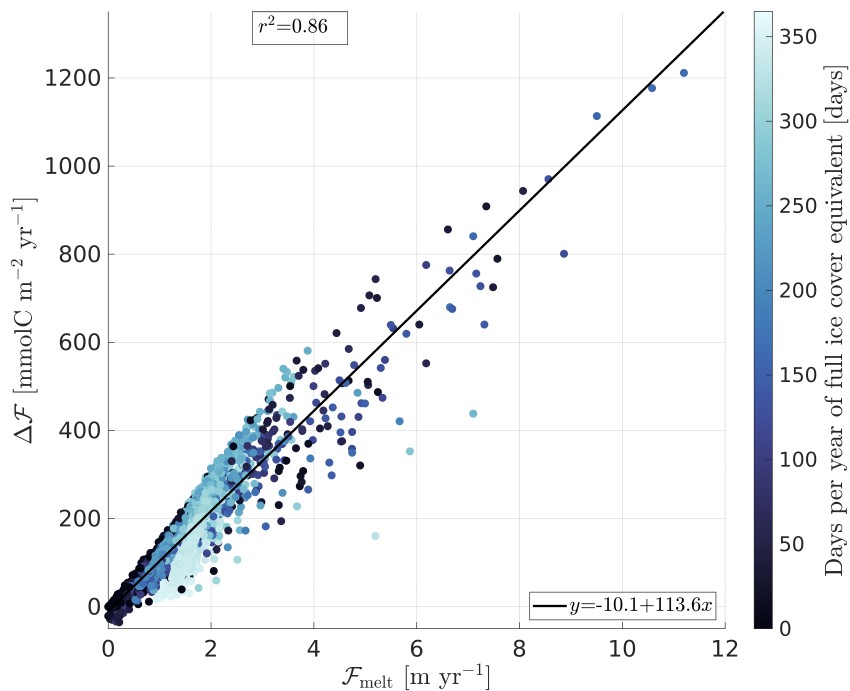

**Figure 4.** Scatter plot of the 1D Arctic-wide runs. The supplementary carbon uptake $\Delta\mathcal{F}$ is plotted as a function of the gross annual ice melt. The color of the dots shows the temporal integral of ice melt over the year, in days. The squared correlation coefficient $r^2$ between both variables is given in the top left corner.

Another driver for $\Delta\mathcal{F}_m$ is the yearly integrated ice concentration (Fig. 4, colors), which is largest where full ice cover persists for most of the year (Fig. 3b). While model experiments with lower ice coverage (dark blue) follow the regression well (solid black line), runs with higher ice coverage (light blue) have a steeper slope.

The 1D simulation ensemble can be used to calculate a yearly Arctic-wide increase due to ice-ocean carbon flux, for the 2014-2019 period. The ICE runs represent an increase of $30.0 \pm 9.1$ % (mean $\pm$ standard deviation calculated over the 6 years period), compared to the CTRL runs. Equation 6 depends on the parameterization of carbon ice-ocean flux and of air-sea CO2 flux, but is not otherwise model-specific. Therefore, it can be applied to other model outputs.

## 4.2 Application to an Earth System Model

The amplification of the air-sea $CO_2$ exchange due to the storage of carbon and alkalinity in ice is sensitive to the gross annual ice melt and the seasonality of the ice concentration. Both parameters are rapidly changing due to global warming. To investigate the impact of these changes on the supplementary carbon uptake, we turned to outputs from the ACCESS-ESM1.5 (Ziehn et al., 2020). This model, as any ESM, does not include the effect of the sea ice carbon pump. We applied the linear



relation in Eq. 6 to estimate the missing carbon uptake of $CO_2$ and, by adding it to the modelled carbon uptake, provide a
corrected estimate of the oceanic carbon uptake in polar regions.

Although the linear relation between gross annual melt and $\Delta\mathcal{F}_m$ (Eq. 6) was derived from daily data, a very similar relationship is obtained when monthly data is used instead (RMSE between daily vs. monthly calculated gross annual ice melt <0.06 m yr$^{-1}$, not shown), giving us confidence that Eq. 6 can be used. The linear relation was applied to the extracted gross ice melt, resulting in a yearly supplementary carbon uptake for each grid cell. Spatially integrated over the area of interest,
this yields an Arctic-wide supplementary carbon uptake due to ice-ocean carbon flux, which can then be added to the model-derived carbon flux over the same area to yield a corrected carbon flux. The ratio between $\Delta\mathcal{F}_m$ and the model-derived carbon flux, expressed as a percentage, allows for easier interpretation of the magnitude of the process. This ratio can be interpreted as a measure of how much the ESM underestimates the Arctic Ocean carbon uptake. Those metrics were integrated over the different periods, yielding cumulative carbon uptake estimates over the historical and projection periods.

Due to the $CO_2$ undersaturation of the Arctic Ocean, the net carbon flux is positive (into the ocean) for all periods and scenarios. During the historical run, the modelled uptake slowly increases from  180 Tg C yr$^{-1}$ in 1850 to  200 Tg C yr$^{-1}$ in 1995 (a linear regression gives a slope of 0.26 Tg C yr$^{-2}$ with $r^2$ = 0.5, $p$-value < 0.001), then stagnates during the last 20 years ($r^2$ = 0.0, $p$-value = 0.4) (Fig. 5a). The supplementary carbon flux, on the other hand, remains relatively constant over the whole period (Fig. 5a), meaning the corrected carbon uptake (Fig. 5a) follows a similar pattern as the model estimate. It
also leads to a slow decrease in the ratio of $\Delta\mathcal{F}_m$ over the model estimate (Fig. 5b). The increase in uptake may be driving increasing $p$CO$_2$ levels in the Arctic Ocean (Ouyang et al., 2020; DeGrandpre et al., 2020).

Projecting into the future, all three climate scenarios show a decrease in modelled and corrected carbon uptakes although interannual variability is high. In scenario SSP5-8.5 (Fig. 5a), and SSP1-2.6 to a lesser extent (Fig. 5a), carbon uptake increases until the 2040s, before dropping rapidly during the remainder of the century. The severe sea ice decline in SSP5-8.5 leads to a
similar decrease in $\Delta\mathcal{F}_m$, while the two other scenarios show a relatively constant $\Delta\mathcal{F}_m$ over the 21st century.

Those scenarios differ in how large the fraction of $\Delta\mathcal{F}_m$ is compared to the total carbon uptake (Fig. 5b). Over the historical period, it slowly decreases starting above 15 % to arrive at around 12.5 % in 2015. It keeps decreasing in SSP5-8.5 to reach 5 % in 2100, but the other scenarios show a different story, levelling off at around 11 % in SSP2-4.5, and returning to 15 % in SSP1-2.6.

Integrated over 1850-2100, the modelled carbon uptake sums up to 41.6, 40.2 and 42.3 Pg C for scenarios SSP1-2.6, SSP2-4.5 and SSP5-8.5 respectively, and the supplementary carbon uptake adds another 5.7, 5.6 and 5.3 Pg C respectively (Table 2). Those cumulative supplementary carbon fluxes represent 12.5 to 14.1 % of the model-derived cumulative flux (Table 2).

Therefore, discarding the storage of carbon in sea ice in ESMs can lead to a significant underestimation of the carbon uptake in the Arctic Ocean, with varying impacts depending on the scenario considered, as described in the Discussion.



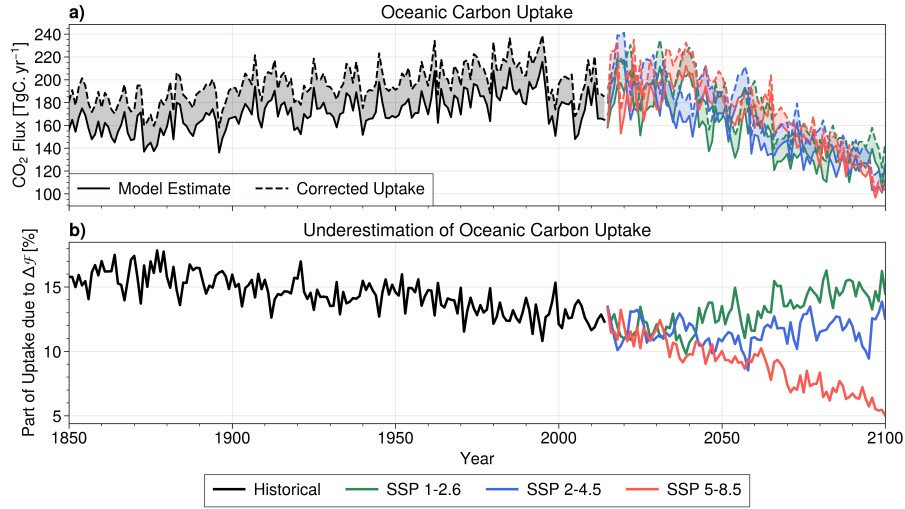

**Figure 5.** Correction of ACCESS-ESM1.5 Arctic Carbon Uptake by applying the linear relation to model outputs, for historical (black), SSP1-2.6 (green), SSP2-4.5 (blue) and SSP5-8.5 (red) scenarios. (a): Arctic oceanic carbon uptake from ACCESS-ESM1.5 (solid lines) and corrected estimates (dashed lines). The shaded area between lines corresponds to the supplementary carbon uptake $\Delta\mathcal{F}_m$. (b): Ratio of $\Delta\mathcal{F}_m$ over model-derived carbon flux, expressed in percentage. This gives an estimate of how much the ACCESS-ESM1.5 model underestimates Arctic oceanic carbon uptake due to the lack of parameterization of ice-ocean carbon flux.

**Table 2.** Cumulative carbon uptake from ACCESS-ESM1-5 model outputs, for historical (black), SSP1-2.6 (green), SSP2-4.5 (blue) and SSP5-8.5 (red) scenarios. Corrected refers to carbon uptake calculation while taking account of sea ice-induced supplementary carbon uptake as calculated using our linear regression (Eq. 6). Percentage refers to the normalized difference (in %) between the model-derived and corrected cumulative carbon flux.

| Cumulative carbon flux (Pg C) | Historical (1850 to 2015) | SSP1-2.6 (1850 to 2100) | SSP2-4.5 (1850 to 2100) | SSP5-8.5 (1850 to 2100) |
| --- | --- | --- | --- | --- |
| Model-derived | 28.3 | 41.6 | 40.2 | 42.3 |
| Corrected | 32.3 | 47.3 | 45.8 | 47.6 |
| Percentage | 14.1 % | 13.7 % | 13.9 % | 12.5 % |

## 5 Discussion

In this study, the link between ice-ocean and air-sea carbon fluxes was investigated using two independent methods: a theoretical framework and numerical modelling. The methods provide consistent, complementary results, both pointing to a linear relationship between $\Delta\mathcal{F}$ and ice melt and an exponential relation with the open-water fraction (Eq. 5 and Fig. 4).



Only three assumptions were made during the theoretical derivation. The assumption of a constant $\frac{\partial pCO_2}{\partial DIC}$ was addressed

in Sect. 2. The second assumption was a constant value of the mixed layer depth $H_0$, also discussed in Sect. 2. The third assumption is the negligible effect of non-linearities in the carbonate system. Here, it is worth noting that the 1D numerical model does not rely on those assumptions and accounts for the varying $\frac{\partial pCO_2}{\partial DIC}$ and $H_0$, and for the non-linearities of the carbonate system. Therefore, the good agreement between the theoretical framework and the model ensemble results builds confidence that these assumptions are justified. Back-of-the-envelope calculations using typical orders of magnitudes ($pCO_2$

= 350 µatm, changes in $pCO_2$ = 20 µatm) also show that non-linearities would represent less than 10 % of the total changes induced by the temperature, salinity, DIC and alkalinity variations, further supporting our assumptions.

A simplified version of the theoretical equation 5 can be evaluated with $g$ and $\frac{\partial pCO_2}{\partial t} \frac{1}{H_0} k_g S_{CO_2}$ considered as constant (cf. Supplementary Materials, Section S1.4), to better compare both methods. The ice concentration and freezing-melting flux used to force the 1D model (described in Section 3.1) can then be applied to this simplified version to calculate $\Delta \mathcal{F}_t$ (Fig. 6). While

the constant values and the offline calculations of $\Delta \mathcal{F}_t$ prevent a quantitative comparison with $\Delta \mathcal{F}_m$ shown in Fig. 5, both methods provide a consistent qualitative behaviour, with a clear linear relationship between $\mathcal{F}_{Melt}$ and $\Delta \mathcal{F}$, and an increasing slope with increasing ice cover.

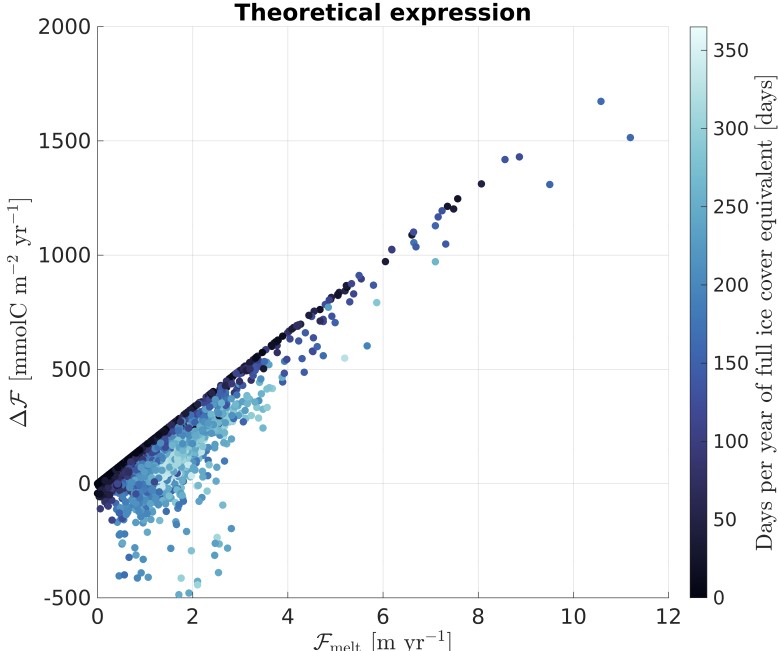

**Figure 6.** Evaluation of a simplified version of Equation 5 with the 1D model forcings: $g(t)$ and $k_g\, S_{CO_2}$ are considered as constant at -315 µatm and 0.073 (cf. Supplementary Materials). The general shape of the scatter plot shows reasonable agreement with the online calculation of the supplementary carbon flux shown in Figure 4.



To interpret the relatively complex equation obtained in the theoretical framework (Eq. 5), we considered that ice formation is associated with ice-covered waters, related to the exponential term. Again, this simplification is supported by results from the NAPA model (cf. Sect. 4.1 and Supplementary Fig. S2). The functional form may not apply to some regions with distinct ice regimes, including ice-exporting polynyas and ice-importing marginal ice zones. In those regimes, the exponential term and therefore the slope of the relation between $\Delta\mathcal{F}$ and ice melt would be different. However, our solution is applicable to most of the Arctic Ocean.

We presented an approach for how Arctic carbon uptake estimates from ESMs can be corrected using our linear relation between $\Delta\mathcal{F}_m$ and sea ice melt (Sect. 4.2). In doing so, past and potential future impacts of the sea-ice carbon pump in the Arctic can be analyzed. Our analysis suggests that uptake due to the sea-ice carbon pump increased during the historical period (Fig. 5a) due to longer open-water seasons and increased atmospheric $p$CO$_2$. This is consistent with observations in the Canadian Arctic where higher $p$CO$_2$ levels are correlated with low ice extent (DeGrandpre et al., 2020). Because the sea ice carbon pump only applies to the seasonally ice-covered areas, the decline in ice extent translates into a stagnation of the supplementary carbon uptake toward the end of the historical period and decreases during all SSP scenarios (Fig. 5a). In the SSP5-8.5 projection, the inhibition of the impact of carbon storage in sea ice is linked to drastic ice loss and therefore to less ice melt. In SSP1-2.6 and SSP2-4.5, the ice seasonal cycle remains significant, leading to a larger importance of $\Delta\mathcal{F}_m$.

In all scenarios, except SSP5-8.5, we deem the current and future role of carbon storage and release by sea ice as non-negligible. Without it, the ACCESS-ESM-1.5 model could be underestimating carbon uptake over seasonally ice-covered areas by 10 to 15 %. Note that this range differs from our calculation using the shorter NAPA model run from 2014 to 2019, in which the supplementary carbon uptake increases the yearly carbon uptake by 30.0 $\pm$ 9.1 % (mean $\pm$ standard deviation over 6 years). The discrepancy is mostly due to a lower ice melt simulated by the ACCESS model compared to the NAPA model ( 18 % lower), though both models have a reasonable agreement with satellite observations in terms of sea ice extent and concentration. We note that ACCESS-ESM-1.5 is the only CMIP6 model that provided the ice-ocean freshwater flux and air-sea CO$_2$ flux, which are necessary inputs for our parameterization. An extension of this calculation to other ESMs would be possible if suitable output was available for more models.

Our estimated supplementary carbon flux is consistent with numbers given by Rysgaard et al. (2011) who suggested that the sea-ice carbon pump could represent 20 % of the air-sea CO$_2$ flux in open Arctic waters at high latitudes. Our estimates are higher than those from two other modelling studies. Grimm et al. (2016) reported that 7 % of simulated net polar oceanic CO$_2$ uptake is due to the sea ice carbon pump. Moreau et al. (2016) found a weakened Arctic carbon sink when including the sea-ice effect. While a direct comparison with those studies is difficult, we suggest that the vertical resolution is crucial for properly resolving the mechanisms at play. The coarse resolution used by Grimm et al. (2016) and Moreau et al. (2016) (9 and 10 layers in the first 100 m, respectively, compared to 9 layers in the first 10 m in our configuration) prevent them from capturing the shallow summer mixed layer observed in the Arctic. Using the same resolution as Moreau et al. (2016) in our 1D model leads to significant changes in the magnitude of the air-sea flux, either positive or negative depending on whether the mixed-layer depth is under- or over-estimated. The importance of high vertical resolution, capable of properly representing the shallow mixed layers in Arctic regions, is not surprising.



While the amplified seasonal cycle of carbonate properties found in our study agrees well with Mortenson et al. (2020), they suggest a negligible impact of ice-ocean carbon flux on annual oceanic $CO_2$ uptake. A potential source of this disagreement could be their lower alkalinity-to-DIC ratio in sea ice (1.25 in their study, 1.8 for this study's reference case). We have shown that the resulting supplementary carbon uptake is sensitive to this ratio (Sect. 4.1 and Fig. 2).

Our parameterization of the alkalinity-to-DIC ratio as constant may be overly simplistic, as it is known to increase over time. The ratio can change for several reasons: (1) $CO_2$ outgassing from ice to the atmosphere when brine is expelled at the surface (Miller et al., 2011) or when permeability is restored by increasing temperatures in early spring (Delille et al., 2014; Nomura et al., 2010) decreases DIC, although uncertainties in these fluxes are high (Watts et al., 2022), (2) primary production from ice algae consumes $CO_2$ and nitrate, therefore reducing DIC while increasing alkalinity (Delille et al., 2007; Rysgaard et al., 2007), (3) formation of ikaite crystals trapped in sea ice retains alkalinity while $CO_2$-enriched brine is exchanged with seawater (Rysgaard et al., 2007, 2009, 2011). However, the main driver of supplementary carbon uptake is sea ice melt, occurring towards the end of the seasonal cycle, when the alkalinity-to-DIC ratio is expected to be highest (Sect. 4.1). Therefore, applying a constant, high ratio is likely to best match real conditions while keeping the parameterization in its simplest possible form. Moreover, while the value of 1.8 might seem high, it is within the range of observed values (1 to 2, Miller et al., 2011; Rysgaard et al., 2009, 2011). Nonetheless, a better constraint on this ratio is needed, which requires a proper understanding of the conditions of ikaite formation.

The empirical linear relation determined in Sect. 4.1 (Eq. 6) involves annual ice melt only, to the exclusion of ice formation. Outputs from the 3D numerical ice model show that whenever the freeze-melt rate is negative (i.e., ice is forming), the ice concentration is close to 1 preventing gas exchange. While this might be due to artifacts inherent to numerical models (e.g., lack of resolution of small leads), our linear relation is derived for application on the latter and therefore stands in this context. It should be noted that we excluded shallow shelves from our runs, such as the Laptev Sea shelves. Those areas are highly productive with regard to ice formation in polynyas (exceeding 7 meters per year, Dmitrenko et al., 2009) and subject to active leads in winter. Therefore, in those regions, during ice formation, carbon storage in sea ice could yield anomalous outgassing, though intense ice formation has also been linked to enhanced $CO_2$ uptakes (Else et al., 2011). Brine sinking in those areas is also significant enough to form deep water masses and is therefore likely to provide a carbon export mechanism over multiyear time scales. Investigating this mechanism would require a fully coupled 3D model.

In this study, a 1D model was used preferentially for computational reasons. This provided more flexibility for parameterization and sensitivity tests and allowed us to generate a large ensemble of simulations which would be computationally prohibitive with a full 3D model. For the same reason, we disabled the biological processes in our 1D model. It could be hypothesized that respiration will increase $pCO_2$ in winter when ice is acting as a lid and primary production will lower it in summer, in phase with the chemical process described here, thus further amplifying the sea ice carbon pump. The storage and release of carbon by sea ice complete the picture drawn by the rectification hypothesis (Yager et al., 1995) which assumes that half of the air-sea $CO_2$ exchange that would be occurring in the typically ice-free ocean is cancelled by the presence of sea ice. While this rectification hypothesis is fully applicable in areas of local ice formation and melt, the southern-most areas of our domain of interest (e.g., Labrador Current and East Greenland Current, Fig. 3) only involve melting of advected ice, usually in





winter and are therefore out of phase with the previously described seasonal cycle of $p\mathrm{CO}_2$. Melting of advected sea ice would then decrease $p\mathrm{CO}_2$ and increase carbon uptake in winter without modifying it in summer. Deep convection events frequently

happening in those areas could then have important consequences for the carbon export at depth, but this is beyond the scope of this study.

## 6 Conclusions

In this study, we used two independent but consistent approaches, a theoretical framework and numerical models, to explore the effect of storage and release of alkalinity and DIC by sea ice on air-sea $\mathrm{CO}_2$ fluxes. Our theoretical derivation and numerical

results show that the ice-ocean carbon flux amplifies the seasonal cycle of surface $p\mathrm{CO}_2$ in phase with the seasonal cycle of sea ice concentration. This leads to a significant increase of oceanic carbon uptake in seasonally ice-covered areas in the Northern Hemisphere. One of the key findings of this study is that ice melt is a direct driver of the supplementary carbon uptake and can therefore be used to correct carbon uptake estimates. This supplementary carbon uptake accounts for 30 % of Arctic Ocean carbon uptake according to our regional, high-resolution model and for 5 to 17 % in the global, lower-resolution

ACCESS-ESM1.5 model, depending on the chosen scenario.

We also provide two novel relations to estimate the impact of sea ice carbonate on air-sea carbon flux. The first (cf. Eq. 5 for the full expression of $\Delta\mathcal{F}_t$), derived from a theoretical framework, can be useful for analyzing observational datasets and decomposing sources of $p\mathrm{CO}_2$ variability. The second, $\Delta\mathcal{F}_m = 113.6 \cdot \mathcal{F}_{Melt} - 10.1$, derived from a linear regression on numerical data, can be used to estimate the missing supplementary carbon uptake in numerical models that do not account for

the sea ice carbon pump. An important strength of our theoretical framework is that no geographical assumption was made in its derivation. Eq. 5 can therefore be applied to both the Northern and Southern Hemispheres, keeping in mind that alkalinity and DIC values in sea ice may be different between both regions, due to environmental conditions (Delille et al., 2014; Fransson et al., 2011; Rysgaard et al., 2011).

While the results presented here offer a straightforward way for estimating the missing carbon uptake in ESMs, additional

sea ice and under-ice observations will help to better constrain the impact of carbon storage in sea ice onto air-sea fluxes. Furthermore, it seems prudent to add sea ice biogeochemistry to numerical models to reduce uncertainties, especially in regional studies.

This study emphasizes the importance of accounting for carbon storage in sea ice in numerical models for an accurate simulation of carbon fluxes in polar regions. Further model runs explicitly simulating the sea ice carbon pump in projection

scenarios would help validate our results and would provide useful insights into the future carbon cycle in the Arctic and Southern Oceans. Observational constraints on the temporal and spatial variability of the alkalinity-to-DIC ratio in sea ice and a better mechanistic understanding of the fate of brine during ice formation season are crucial for properly simulating those processes. The importance of the sea ice carbon pump should also be kept in mind when estimating fluxes from observations. A better accounting of the sea ice carbon pump will also facilitate the global effort to better constrain the carbon cycle in the

oceans and to understand its changes under climate change.



*Data availability.* The 1D model outputs are available at https://doi.org/10.5281/zenodo.7038942. The ACCESS-ESM1.5 data can be accessed at https://esgf-node.llnl.gov/search/cmip6/ (last access: 31 August 2022; Ziehn et al., 2020). The BGOS Mooring In Situ pCO2 and pH time-series are provided by Michael DeGrandpre and hosted by the Arctic Data Center, https://doi.org/10.18739/A28C9R46N (last access: 31 August 2022; DeGrandpre et al., 2019).

*Author contributions.* BR, KF and EO conceived the study. BR carried out 1D model simulations, validation, and analyses. TB, XH and YL assisted with setup and validation of the NAPA model. TB assisted with ACCESS-ESM1.5 data access and analysis. BR, KF, EO and MDD discussed the results. BR, KF and EO wrote the paper with contributions from the coauthors.

*Competing interests.* The authors declare that they have no conflict of interest.

*Acknowledgements.* BR was supported by an Ocean Frontier Institute Grant to KF and EO, and the NSERC ACCSC project "Quantifying
and Predicting Canada's Marine Carbon Sink". MDD was supported by U.S. NSF Office of Polar Programs grants OPP-1723308 and OPP-1951294. TB received funding from the Research Council of Norway (RCN) under grant No. 275268 (COLUMBIA) and from the European Union's Horizon 2020 research and innovation programme under grant agreement No. 820989 (COMFORT). The work reflects only the authors' view; the European Commission and their executive agency are not responsible for any use that may be made of the information the work contains. We acknowledge the World Climate Research Programme, which, through its Working Group on Coupled Modelling,
coordinated and promoted CMIP. We thank the CSIRO for producing and making available the ACCESS-ESM1.5 model output, the Earth System Grid Federation (ESGF) for archiving the data and providing access, and the multiple funding agencies who support CMIP and ESGF. The BGOS mooring in-situ pCO2 time-series data used in this study for validation are available through the U.S. National Science Foundation (NSF) Arctic Data Center(https://arcticdata.io). BR would like to acknowledge NSF support for participating in the BGOS/JOIS 2021 cruise.





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
