# Peer review of "Underestimation of oceanic carbon uptake in the Arctic Ocean: Ice melt as predictor of the sea ice carbon pump"

_EGUsphere, 2022_

## Author Comment (AC2)

Below we have pasted the Reviewer's comments in their entirety interspersed with our responses in blue font.

**Comments from Reviewer 1:**

We are grateful for the Reviewer's comments and suggestions, and are glad that they enjoyed reading this manuscript. In our revision, we have addressed the Reviewer's comments and suggestions as described below and hope we have clarified the points raised.

The manuscript by Richaud et al., "Underestimation of oceanic carbon uptake in the Arctic Ocean: Ice melt as predictor of the sea ice carbon pump" investigate how the oceanic carbon uptake is strongly modulated by sea ice. They base their work on previous studies showing that the ratio of alkalinity to dissolved inorganic carbon in sea ice is higher than in the underlying water and previous suggestions that this storage amplifies the seasonal cycle of sea water $pCO_2$ and leads to increased carbon uptake in the ocean. They have two independent approached; a theoretical framework and a simple parameterization of carbon storage in sea ice in a 1D physical-biogeochemical ocean model. Sensitivity simulations show a linear relationship between ice melt and an amplified seasonal carbon uptake. In addition, they estimate a 30% increase in carbon uptake in the Arctic Ocean compared with no ice amplification. Applying this ice melt parameterization to future scenarios of an Earth System Model suggest that the Arctic Ocean carbon uptake is underestimated by 5 to 15%.

Overall comment:
The paper provides new and valuable results for our understanding of the biogeochemical processes in sea ice and how sea ice modulate the air to ocean carbon transfer in the Arctic Ocean and ice covered seas. The paper is well structured, well written and the results highly interesting to a broader scientific audience interested in global warming. Therefore, I will recommend the publication of this work if the authors consider the minor comments below.

**Response:** We appreciate the positive assessment.

Specific comments:
Line 37. Suggest to provide an additional reference to Rysgaard et al. 2013 (doi:10.5194/tc-7-707-2013) where the link between ikaite crystals trapped within the sea ice matrix and the distribution of alkalinity are shown for winter ice conditions.

**Response:** Agree. Done.

Line 42. After DIC ratio, I suggest to provide a reference to Rysgaard et al. 2012 (doi:10.5194/tc-6-901-2012) where ikaite dissolution is shown for melting sea ice and how this affect $pCO_2$ and pH levels in Arctic surface waters.

**Response:** Agree. Done.

Line 89. I'm not sure DIC and alkalinity are homogeneous in sea ice. They are probably more C shaped. However, it is a fair assumption considering the few existing observations in different forms of sea ice.

**Response:** We absolutely agree, alkalinity and DIC are likely to be vertically variable. However, their homogeneity is a necessary assumption to analytically derive and solve the differential equation. Furthermore, if values are used that are representative of the freezing and melting ice over a seasonal cycle, we believe the assumption is reasonable. We added a brief discussion of this assumption in the manuscript (l. 402):

*"[Our parameterization of the alkalinity-to-DIC ratio] may be overly simplistic. First, the vertical profiles of alkalinity and DIC in sea ice, assumed homogeneous here, might be C-shaped to follow salinity profiles, though observations do not necessarily support a vertical heterogeneity (e.g. Miller et al., 2011 ; Rysgaard et al., 2009). As long as the parametrized values are representative of the freezing and melting ice over a seasonal cycle, we believe that the vertical homogeneity assumption is reasonable. Second, the alkalinity-to-DIC ratio is known to increase over time.[...]"*

Line 188-190. I am surprised the biological terms had a negligible impact on carbon uptake. Could you elaborate a little more why that is?

**Response:** We understand the confusion and found this to be a difficult point to explain concisely. Note that we do not mean to say that biology has a negligible impact on carbon uptake in general, but only on the **supplementary** carbon uptake due to the presence of alkalinity and DIC in sea ice. In other words, calculating the difference between a CTRL run without biology and the corresponding ICE run without biology yields a similar supplementary carbon uptake as the difference between the CTRL run and ICE run with biology. The supplementary carbon uptake is not driven by biological processes but by the chemical properties of ice and sea water.

We have attempted to clarify this by adding the following text (l. 199):
 *"[the biological terms have a] similar impact on carbon uptake regardless of whether the carbonate system inside sea ice is represented or not, and thus yield a [negligible impact on supplementary carbon uptake]"*

Line 325. The assumption of a constant mixed layer is a good beginning. However, I expect leeds and polynyas (ice fabrics) could elevate the carbon uptake. I'm aware that this will require very high-resolution modelling, but could be very interesting thing to look into after your present work. Looking forward to a follow up study later.

**Response:** Absolutely agree, the spatial heterogeneity of sea ice concentration and mixed layer conditions would be interesting to investigate. Several papers have observed intense carbon uptake in leads (e.g. Else et al., 2011, https://doi.org/10.1029/2010JC006760). But as mentioned by the Reviewer, the model requirements for such a study are completely different from what we have used here.

Line 355. Here you state that models without the ice pump parametrization may underestimate carbon uptake over seasonally ice-covered areas by 10-15%. In the abstract this number is 5 to 15 %.

**Response:** Indeed. When reporting this 10-15% number, we exclude the scenario SSP8-5.5, but we included it in the abstract numbers to be conservative. We have clarified this in the manuscript (l. 370):

*"Without it, the ACCESS-ESM-1.5 model could be underestimating carbon uptake over seasonally ice-covered areas by 5 to 15 %, **or 10 to 15 % if we exclude SSP5-8.5**."*

Line 360. I'm happy to see that your estimated supplementary carbon flux is consistent with numbers provided by Rysgaard et al 2011. Do your model also include the Southern hemisphere and would it be possible to provide a number for sea ice Antarctica? Could be a really interesting follow up study after this work.

**Response:** It would indeed be interesting to look at the Southern Ocean. Conditions there are significantly different and might provide a different linear relationship with ice melt. Unfortunately, our forcing data set does not include the Southern hemisphere, preventing us from easily extending the analysis.

Line 370. Your statement regarding the importance of high vertical resolution in the model to represent the shallow mixed layer is an important one. In order for the carbon pump to work, the CO2 released from ikaite production in sea ice only has to go below a thin mixed layer to prevent (or greatly reduce) exchange with the atmosphere in the Arctic Ocean due to an impermeable sea ice cover (autumn, winter and spring). As this cold water below the mixed layer meets warmer and saltier Atlantic water on its way out of the Arctic Ocean it will sink in the Denmark Strait. At the same time melting sea ice in the summer will be in contact with the atmosphere and result in dissolution of ikaite and release of excess alkalinity to surface waters and hereby stimulate CO2 uptake from the atmosphere. Could be interesting to look into regional differences in air-ocean CO2 uptake.

**Response:** That is our understanding as well. Moreover, the vertical resolution conditions the resolution of the mixed layer, which in turn has an important impact on the volume in which the ikaite can dissolve in summer and therefore on the lowering of $p$CO$_2$ and strength of the supplementary carbon uptake. We have emphasized this in the manuscript by adding (l. 465):

*"A high vertical resolution would be crucial to properly resolve the shallow Arctic summer surface mixed layer and the carbon subduction."*

Line 395. Polynyas and leads. Interesting and I would love to see more on this modelling in the future.

**Response**: We appreciate this comment and, following the previous addition, we now mention those studies explicitly (l. 467):

*"Modelling studies dedicated to leads and polynyas would also help to qualify and quantify the sea ice carbon pump in those areas of intense mixing, as well as providing guidelines on how to parametrize those mesoscale ice features in low resolution ESMs."*

Summary: I really enjoyed reading this study.

**Response**: We really appreciate this comment and are grateful to the Reviewer.

---

## Author Comment (AC3)

Below we have pasted the Reviewer's comments in their entirety interspersed with our responses in blue font.

**Comments from Reviewer 2:**

We appreciate the Reviewer's comments and agree that our manuscript was lacking a proper consideration and discussion of the two key questions he raised. We have addressed those points in our revision as detailed below and believe the revised manuscript is clearer and significantly improved as a result.

**General comment**

This is a study on the effect of sea ice on the carbon exchanges in ice-covered seas, based on a rather conceptual approach and an offline ESM application. I find the paper reasonably well presented, yet insufficiently developed and subject to important methodological ambiguities.

The origin of the latter could be that the paper has not taken full benefit of the literature, ignoring key processes (subduction of carbon below the mixed layer) and progresses in the definition of the sea ice carbon pump.

There is space for a conceptual study of the seasonal cycle of air-sea carbon exchanges in ice-covered seas, however important aspects (including basic calculations and experimental design for model experiments) would need to be reworked, in my opinion, before the paper represents significant progress against state-of-the-art.

**More detailed comments**

The key question, as presented in the abstract « how the storage of carbon in sea ice affects air-sea CO2 flux and quantify its dependence on the ratio of TA vs DIC in ice », was central to two contributions on the topic by Grimm et al (2016) and Moreau et al (2016).

Both studies are cited in the submitted paper, however two key progresses that these made were not considered in the submission.

First, both author teams conclude that the subducted fraction of the carbon anomaly generated in the mixed layer during sea ice growth is a key player in the intensity of the sea ice carbon pump (SICP). Rysgaard et al assumed a 100%-efficiency of carbon subduction below the mixed layer (leading to maximum efficiency), whereas both Moreau et al and Grimm et al suggest the subduction efficiency could be less than 5%, which would drastically decrease the intensity of the SICP compared to the large numbers of Rysgaard et al 2011. This finding is not mentionned and not accounted for in the conceptual model presented in the submitted paper, which **I find problematic as subduction efficiency is a key player and source of uncertainty**.

**Response:** We agree with the Reviewer that subduction efficiency is a key player for determining the ultimate fate of atmospheric carbon taken up by the ocean and its long-term storage, because subduction provides a mechanism for transport of carbon into the deeper ocean, out of reach from atmospheric influence. We now state this more clearly in the manuscript (l. 44) where we have added:

*"and the long-term fate of the carbon taken up is controlled by subduction processes, including advection of water masses to depth (Bopp et al., 2015; Karleskind et al., 2011)."*

However, an investigation of subduction is beyond the intended scope of this study, which addresses the effect of the sea ice carbon pump on the air-sea flux (as stated in the abstract) and considers seasonal to annual time scales. We are confident that the efficiency of subduction is not a strong determinant of the influence of carbon storage in sea ice on air-sea flux. We further note that the current Arctic Ocean is **strongly undersaturated** in $CO_2$ with regard to the atmosphere. Given the undersaturated state of the Arctic Ocean, even if subduction was nonexistent, the sea ice carbon pump would still affect air-sea flux on the seasonal to annual time scales we are considering. Indeed, by amplifying the $p\mathrm{CO}_2$ seasonal cycle in phase with the ice concentration, the sea ice carbon pump generates a higher seasonal uptake than without the carbon storage in sea ice, even if its mean value were to remain constant or increase slightly (i.e. if subduction was inefficient). This would potentially mean a faster "catch-up" of the surface Arctic Ocean $p\mathrm{CO}_2$ values toward atmospheric values if subduction is as low as suggested by Moreau et al. and Grimm et al.

As shown in our study, the choice of vertical resolution impacts model-derived estimates of the supplementary carbon uptake; we believe that the relatively coarse resolution used by Moreau et al. and Grimm et al. raises the questions about uncertainties in their estimates about subduction.

In the case of the ACCESS-ESM1.5 model, subduction processes are accounted for in the initial outputs, but an estimate of how this subduction would act on the supplementary carbon pump would require a full run with the ESM with a sea ice biogeochemical component (we note that we do not have access to the ESM model). Our methodology inherently assumes that all the supplementary carbon is subducted, which we didn't mention in the initial manuscript. We have corrected this omission as follows (l. 304):

*While subduction processes are simulated in the initial outputs, our offline methodology does not correct mixing and advective carbon transport for the supplementary carbon due to the sea ice carbon pump. Therefore, an inherent assumption to our methodology is the subduction of all the added carbon."*

Since the physical circulation remains unchanged by the sea ice carbon pump, we could expect mixing and advective processes to be linearly dependent on the DIC and alkalinity gradients, hinting towards a proportional increase of the additional carbon uptake by subductive processes, but this remains to be proven. We have introduced another sentence in the manuscript to briefly discuss the ACCESS-ESM1.5 case, at l. 393:

*"The output from the ACCESS-ESM1.5 model account for subduction, but the fate of supplementary carbon estimated here cannot be determined without running a new configuration of the model. It is therefore unknown whether carbon flux driven by advection and mixing would proportionally increase and export the supplementary carbon or whether the latter would saturate the surface mixed layer, leading seawater $p\mathrm{CO}_2$ to catch-up with atmospheric values faster than without accounting for the sea ice carbon pump."*

We believe that the sea ice carbon pump is an important enough process that deserves investigation and probably shouldn't be neglected in ESMs as it currently is. We acknowledge that the points raised by the Reviewer should have been clearer and, in addition to the changes described above, have added the following sentences:

- l. 388: *"On top of that, a proper representation of subduction, included in the Grimm and Moreau studies but beyond the scope of the present one-dimensional study, would be important to more fully understand the long-term fate of carbon in the global ocean. Yet, in an undersaturated ocean, the amplification of the $pCO_2$ seasonal cycle can in itself explain an increased seasonal carbon uptake. Without any subduction, this would then lead the Arctic Ocean to reach equilibrium with the atmosphere faster than without accounting for the sea ice carbon pump, eventually saturating the surface ocean and reducing the carbon uptake."*

- l. 465: *"including the role of mixing and advective processes on the fate of the added carbon. A high vertical resolution would be crucial to properly resolve the shallow Arctic summer surface mixed layer and the carbon subduction"*

Second, Moreau et al and Grimm et al decomposed the sea ice carbon pump into several sub-components. The decomposition of the SICP has important implications on the requirements for what the reference control case should be in model experiments, in order to draw proper conclusions on the sea ice carbon pump. Both aforementioned studie acknowledge that the SICP results from mostly three groups of processes. (i) Sea ice growth, which implies an uptake of freshwater from the ocean to the ice, (ii) brine rejection, which proportionally decreases the uptake of solutes in sea ice, and (iii) active biogeochemical processes, which modify the TA/DIC ratio in sea ice.

**Response:** We appreciate this comment which has prompted us to state the assumptions in our model experiments more clearly and contrast them directly with the assumptions in the previous publications. We have added the following sentences:

- l. 57: *"The sea ice carbon pump is considered to result mostly from three groups of processes: (i) sea ice growth or melt, which implies a freshwater flux (upward or downward) from the ocean to the ice, (ii) brine rejection, which proportionally decreases the uptake of solutes in sea ice, and (iii) active biogeochemical processes, which modify the alkalinity to DIC ratio in sea ice. Most, if not all, Earth System Models (ESMs) lack a representation of biogeochemical processes within sea ice and are therefore unable to account for (ii) and (iii), but encompass (i) by dilution and concentration of tracers similar to the handling of precipitation and evaporation. In the present study, we do not distinguish between (ii) and (iii) and instead consider that the carbon cycling in sea ice encompasses both aspects. We also consider our reference point (later referred to as CTRL) to be that of current ESMs, i.e., they include processes (i) but not processes (ii) and (iii)."*

- l. 78 (additions in bold): *"[The supplementary carbon flux] is quantified here as the difference in air-sea $CO_2$ flux between a* **reference** *situation where there is no ice-ocean carbon flux,* **i.e including aforementioned processes (i) but not (ii) nor (iii),** *and situations where ice-ocean carbon flux occurs* **, i.e including (i), (ii) and (iii)."**

In this context, depending on the question of interest, the reference point for « no sea ice» could be several things, and basically two points of view are used in the literature. A first point of view (#1) is to assume no freshwater and no solute uptake/release associated with sea ice, wherease point of view #2 assumes no solute uptake, but some freshwater uptake associated with sea ice. #1 (no fw and no solute uptake) was generally adopted by Moreau et al and Grimm et al and implies that there is virtually no change in surface concentrations due to sea ice at all in their CTRL experiment. #2 assumes full rejection of solutes and would lead to the strongest effect of sea ice on surface concentrations.

**Response:** We agree with this description. Moreau et al. conducted three different experiments: CTRL which corresponds to no freshwater and solute uptake (point of view #1), PHYS which has freshwater uptake as well as some solute uptake proportional to salinity (so doesn't really match point of view #2), and CARB which includes a modified alkalinity to DIC ratio. They then compare PHYS – CTRL, and CARB – PHYS. The approach adopted in our study is closer to this latter experiment, although it doesn't perfectly match it.

Grimm et al. have a similar distinction, with their "control" run matching point of view #1 and their R0 experiments corresponding to point of view #2. Our CTRL experiment matches R0, instead of their "control" experiment, which could potentially lead to confusion.

**Our rationale is that we wish to use the current state of the art in ESM as the reference point.**

We have attempted to clarify this by adding the following clarifications to the manuscript, on top of the changes described above:

- l. 112: *"[and a control (CTRL) case, where storage is not considered]* **and ice growth or melt only leads to a freshwater exchange**"
- l. 197: *"[Phys includes the advective and dispersive transport terms]* **as well as dilution and concentration due to sea ice melting and freezing or due to precipitation and evaporation**".
- l. 225: *"In both configurations, sea ice growth and melt generates a freshwater flux that concentrates or dilutes tracers at the surface ocean."*

Both points of view can be justified depending on what one is looking for. However, the issue with the submitted study is that, in my understanding, points of view #1 and #2 are adopted in different parts of the paper without notice, which I view as problematic because the conclusions drawn are be completely different if one or the other point of view is adopted.

**Response:** Actually, we have consistently adopted point of view #2. Please cf. below.

Section 2 (conceptual study) assumes point of view #2. In this context, comparing CTRL and ICE differentiates between the effects of full rejection of solutes minus brine rejection (which I find not so useful scientifically, but possibly useful as a technical stage meant to fully decompose things).

**Response:** We agree.

In Section 3 (1D model), I'm not really sure which point of view is adopted. Table 1 says TA=DIC=0 (point of view #1), whereas the text says no ice-ocean carbon flux (line 235), which would correspond to point of view #2. As far as I understand it, the default approach in PISCES assumes point of view #2. Finally, regarding the ACCESS-ESM application, where the analytic perturbation is applied, there is no

mention of what is assumed in that ESM for TA and DIC concentrations in sea ice, hence we don't know the point of view adopted. Clearly, applying the perturbation would require point of view #1, for consistency with the analytical calculations. However, it could well be that model developers would have adopted point of view #2 (I know of many ESMs which adopt point of view #2). In that case, the ESM outputs and the diagnosed perturbation on carbon fluxes would be incompatible, and any finding based on their association would be wrong, I'm afraid.

**Response:** In this section as well, we adopt point of view #2, for the exact reason mentioned by the Reviewer: we need to be comparable with ESMs, which account for tracer dilution and concentration due to sea ice melt and growth but lack a sea ice biogeochemical component. We argue for the need for the modelling community to include BGC in sea ice, so we make sure our reference point matches the current state of ESMs, hence the discrepancy between our CTRL reference and Grimm et al. and Moreau et al.'s. We hope some of the previously mentioned revisions (e.g. l. 57) clarify this perspective.

The values given in Table 1 reflect values of DIC and TA in ice: point of view #1 would require TA and DIC in ice to have the same values as in seawater, which is not the case here: we have TA=DIC=0 in ice, which means no ice-ocean carbon flux (cf. equations line 192-193) but does not prevent dilution or concentration of tracers due to freshwater fluxes.

We acknowledge that this was not explained precisely enough in the manuscript and hope that the new version is clearer, with some of the previously mentioned changes and the following addition:

- l. 301: "[*This model, as any ESM, does not include*] **any carbon storage in sea ice, although the freshwater flux between the ocean and sea ice is accounted for**"

**This ambiguity on the relevant reference point of view cast doubt (on my side at least) on the findings presented.** Properly defining the sea ice carbon pump, translating that into a fit-for-purpose experimental design in all sections of the paper, and making appropriate connexions to concluding statements on the sea ice carbon pump should be clarified before anything else, I reckon.

**Response:** Again, we appreciate the comments and believe the manuscript has benefited greatly from the resulting clarifications and revisions.

---

## Author Response (AR2)

Below we have pasted the Editor's and Reviewers' comments in their entirety interspersed with our responses in blue font.

**Editor Comments:**

Dear Authors,

We have now received both second round of comments from the reviewers. One of the referee was more critical in his first round of comments. You can now see that he would accept the paper with "minor revision", basically stating more clearly what is your position on the importance of subduction in the mixed layer (and below) at the time scale you are working, and why you think this is a reasonable assumption (the Rysgaard vs. Moreau/Grimm conundrum).

Could you please amend your manuscript along those lines, and send me a revised version, both "unaltered" and with "annotated" changes, I will then make a final decision,

I think we are getting there, apologies again for the delays...

Best regards,

Jean-Louis Tison

Dear Dr. Tison,

Thank you for giving us the opportunity to clarify our stance on subduction. We recognize that we were lacking a clear description of how we deal with it and are hopeful that this modified version fills the gap.

Best,
Benjamin Richaud on behalf of all authors

**Comments from Reviewer 1:**

Im happy with the reviewers comments, changes and answer to my initial review.

**Response**: Thank you!

**Comments from Reviewer 2:**

Dear authors,

Thanks for reading my comments, and sorry I took so long to come back (all my fault, but too much work over the last two months).

**Response**: No worries, the last few months were busy on our side as well, so we perfectly understand!

I think a very short amount of extra-work related to subduction.

To rebut my argument on subduction, you say you are focussed on air-sea fluxes at seasonal to inter-annual time scales, whereas some of your plots cover 150 years, and your conclusions on carbon uptake (in the abstract) probably regard integral processes that involve the full carbon budget in the mixed layer... I did not find that very convincing, but I think that was not really addressing what I had in mind (and possibly did not express very clearly).

**Response**: We agree with the reviewer that there is a limitation to our study with regard to the consideration of subduction. As previously stated, on the seasonal time scale, we believe our parametrization is justified. However, when making inferences over the 150 year period, our estimate should be considered an upper bound and we have tried to clearly state this in the discussion as follows (l. 398; added text in bold font):

> "The output from the ACCESS-ESM1.5 model accounts for subduction, but the fate of supplementary carbon estimated here cannot be determined without a proper coupling of a sea ice biogeochemical component. It is therefore unknown whether, **at the decadal time scales considered for that model,** carbon flux driven by advection and mixing would proportionally increase and export the supplementary carbon or whether the latter would saturate the surface mixed layer, leading seawater $p$CO$_2$ to catch-up with atmospheric values faster than without accounting for the sea ice carbon pump. **Thus, our estimate should be considered an upper bound of the impact of the sea ice carbon pump.**"

I think what Moreau & Grimm show is that subduction is very small contributor at short time scales, whereas it is assumed 100% efficient in Rysgaard model. Therefore, what I missed in your model description, is a clear statement on what you do on this assumption, hoping that you are close to neglecting this (which would be fine even on decadal time scales)... Are you on Rysgaard's line (assuming full subduction), or on Moreau/Grimm's line (weak net subduction) ?

**Response**: We do not follow Rysgaard's assumption of full subduction. We believe this is now clearly stated in both the methods where we added the following sentence (l. 223):

> "As a consequence, subduction processes are mostly but not entirely neglected in this 1D model."

and in the discussion with the following edits (l. 379, new text in bold):

> "Our estimated supplementary carbon flux is consistent with numbers given by Rysgaard et al. (2011) who suggested that the sea-ice carbon pump could represent 20 % of the air-sea CO$_2$ flux in open Arctic waters at high latitudes. **Rysgaard et al. (2011) assumed complete subduction of the brine, while we did not**. Our estimates are higher than those from two other modelling studies. Grimm et al. (2016) reported that 7 % of simulated net polar oceanic CO2 uptake is due to the sea ice carbon pump. Moreau et al. (2016) found a weakened Arctic carbon sink when including the sea-ice effect. **Neither of these two studies assumed complete subduction and rather diagnosed it from their model, finding it to be relatively small. It has been previously suggested that the differences between the estimates of Rysgaard et al. (2011) and Moreau et al. (2016) are due to the different assumption about subduction. This study does not support that interpretation.** While a direct comparison **between all**  those studies is difficult, we suggest that the vertical resolution is crucial for properly resolving the mechanisms at play."

With a proper statement on that matters I think your paper could go.

All the best,

Martin Vancoppenolle

Thank you, we are grateful for the comments and believe the manuscript is improved thanks to those.